

# On-Orbit Calibration and Performance Validation of the Yunyao Polarimetric Radio Occultation System

Liang Kan[1], Fenghui Li[2,4], Naifeng Fu[3,4,5], Yan Cheng[3,4], Sai Xia[4], Bobo Xu[4]

[1]School of Electrical and Information Engineering, Tianjin University, Tianjin 300072, China
[2]Department of Earh System Science, Tsinghua University, Beijing 100084, China
[3]School of Marine Science and Technology, Tianjin University, Tianjin 300072, China
[4]Tianjin Yunyao Aerospace Technology Co., Ltd., Tianjin 300300, China
[5]Satellite Technology and Research (STAR) Centre, National University of Singapore, Singapore 119077, Singapore

**Correspondence to:** Liang Kan (kllara@163.com)

**Abstract.** Polarimetric radio occultation (PRO) extends the capability of standard radio occultation (RO) by providing not only the conventional thermodynamic profiles but also information on clouds and precipitation. In early 2025, Yunyao Aerospace Technology Co., Ltd. successfully launched the first Chinese low-Earth-orbit satellite equipped with a PRO payload, generating over 500 measurements per day. Based on this mission, we established an end-to-end PRO data processing chain tailored for operational applications and analysed approximately 53,000 events collected between March 15   and June 2025, in conjunction with the Integrated Multi-satellite Retrievals for Global Precipitation Measurement (GPM) precipitation product (IMERG). The results show that the differential phase ($\Delta\Phi$) remains close to zero under non-precipitating conditions but exhibits distinct peaks at 3-5 km altitude when traversing precipitation layers, with amplitudes strongly correlated with path-averaged rainfall rates. Thresholds of 1, 2, and 5mm h$^{-1}$ are proposed as indicators of precipitation sensitivity, detection confidence, and heavy-rain events, respectively, and a $\Delta\Phi$-to-rainfall intensity mapping 20   table is derived to quantify this relationship. Yunyao PRO data preserve the thermodynamic retrieval quality of conventional RO while enabling effective precipitation detection, thereby providing important data support for the theoretical, technical and data research on the transition of meteorological observations from "temperature, humidity and pressure" observations to new types of observations such as precipitation.

## 1 Introduction

Precipitation events exert long-term and widespread impacts on human society, particularly in the context of intense rainfall. Sudden episodes of high-intensity precipitation and persistent extreme wet spells pose substantial threats to human health and critical infrastructure, while also triggering secondary hazards such as flash floods, landslides, and urban waterlogging. The mesoscale processes that govern the formation and evolution of precipitation are fundamentally controlled by atmospheric convection: upward motions transport near-surface water vapor into the upper troposphere, where decreasing 30   temperatures induce condensation, freezing, and coalescence. These microphysical transformations ultimately manifest as



various hydrometeors (e.g., liquid raindrops, supercooled droplets, ice crystals, graupel, and snow), undergoing phase changes and sedimentation that result in precipitation at the surface. Despite notable advances in convective theory and numerical modeling over the past decades, significant uncertainties remain regarding convective initiation, organizational patterns, microphysical processes, and land–atmosphere interactions. These uncertainties continue to constrain the ability of
operational forecasting systems and climate models to accurately characterize the spatiotemporal distribution, intensity, and persistence of precipitation.

Standard RO retrieves scalar refractivity profiles for temperature, pressure, and water vapor, with the advantages of all-weather capability, high vertical resolution, and global coverage (Spilker et al., 1996; Jin et al., 2014; Teunissen et al, 2017). PRO further exploits electromagnetic anisotropy induced by non-spherical, oriented hydrometeors, with $\Delta\Phi$ reflecting rain
rate, particle shape, orientation, and phase state (Cardellach et al, 2014), as shown in Fig. 1. Operational techniques capable of systematically resolving the vertical structure of convection remain lacking, limiting our understanding of the three-dimensional evolution of water vapor and hydrometeors during precipitation (Cardellach et al, 2010). Passive microwave radiometers (e.g., the Advanced Technology Microwave Sounder, ATMS) are sensitive to upper-level ice clouds and provide clues to cloud–ice processes, while imaging radiometers (e.g., the Global Precipitation Measurement Microwave Imager,
GMI) retrieve column-integrated quantities such as total precipitable water and cloud liquid water. Yet, passive observations suffer from weak vertical resolution, strong dependence on prior information, and non-unique emission–scattering signals. Active precipitation radars (e.g., the Dual-frequency Precipitation Radar, DPR) infer microphysical profiles from backscattering, but cannot separate water vapor from hydrometeors and are strongly affected by absorption and attenuation (An et al., 2019). Furthermore, the limited spatiotemporal coverage of low Earth orbit sensors restricts continuous
monitoring of rapidly evolving convection. The PRO extends GNSS RO for precipitation-sensitive sensing by measuring the $\Delta\Phi$ (Cardellach et al, 2018). This polarization-resolved information enhances sensitivity to cloud–precipitation microphysics and convective organization.

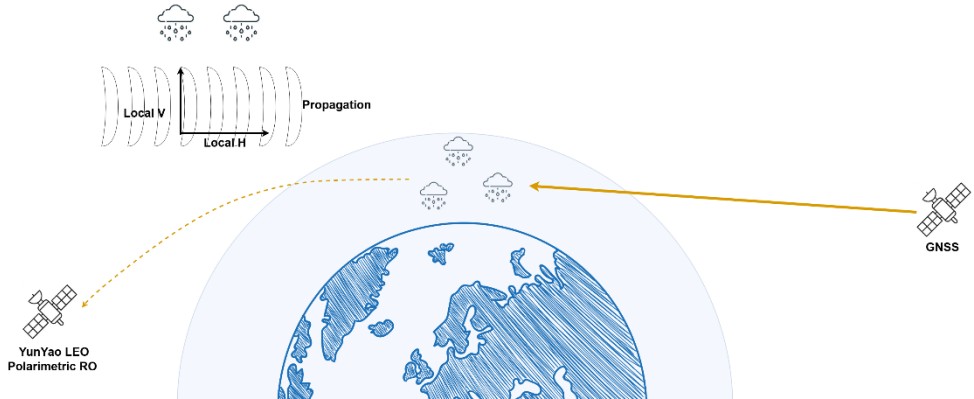

**Figure 1. Illustration of the PRO concept.**



With respect to instrumentation and methodology, Cardellach, Padullés et al. have systematically documented PRO antenna designs, receiver chains, calibration strategies, and retrieval algorithms, and clarified how signal-to-noise ratio, geometric configuration, and frequency plan govern $\Delta\Phi$ detection sensitivity (Padullés et al., 2018). Theoretical analyses and early in-orbit experiments show that $\Delta\Phi$ during precipitation markedly exceeds the expected noise floor, and that the differential-measurement design helps suppress certain systematic errors and ionospheric common-mode residuals. The

Spanish PAZ satellite, launched on 22 February 2018, began radio-occultation and heavy-precipitation experiments on 10 May 2018, acquiring on the order of ~300 GPS-based profiles per day and confirming PRO's pronounced sensitivity to precipitation (Cardellach et al., 2019). At the mechanistic and application levels, Turk et al. revealed correlations between the vertical structure of $\Delta\Phi$ profiles in convective systems and lower-tropospheric moisture profiles—implying diagnostic value for moisture–hydrometeor coupling in deep convection (Turk et al. 2019)—while Padullés et al. (2023) attributed

strong $\Delta\Phi$ peaks to electromagnetic waves traversing layers of oriented frozen hydrometeors (e.g., ice crystals and aggregates). Paz et al. (2024a) used Next Generation Weather Radars (NEXRAD) data to compare the specific differential phase with $\Delta\Phi$ from PRO observations, and found good agreement on the PRO $\Delta\Phi$ and co-located ground-based NEXRAD radars. Retrieving Level-2 variables along the rays remains challenging due to geometric effects—long path integration and vertical superposition—which induce non-uniqueness: different configurations of particle location, amount, and type along

the ray path can produce the same observable (integrated polarimetric phase shift). Lookup-table retrievals were proposed and validated with synthetic experiments (Cardellach et al., 2018). It is worth noting that POR does not degrade traditional thermodynamics measurements (Paz et al. 2024b).

     To fully exploit PRO's spatiotemporal sampling and microphysical sensitivity, Turk et al. (2022) proposed deploying small, tightly clustered PRO constellations to retrieve quasi-simultaneous water-vapor fields for diagnosing inter-model

differences in the precipitation–moisture relationship, and they further examined synergistic retrieval frameworks that combine PRO with microwave imagers/sounders to quantify PRO's marginal contribution to cloud-ice characterization and forecast gain. Because PRO entails only limited hardware and processing-chain modifications to conventional RO payloads, it represents a cost-effective, technically continuous, and scalable evolution of GNSS radio occultation (Turk et al., 2024). Recent demonstrations include the ESA–Spire–Spanish research team integration of a novel triplet polarimetric antenna on

the LEMUR-2 platform, validated on orbit in 2023 (Talpe et al., 2025), and PlanetiQ's launch of its first PRO satellite on 16 August 2024 (Kursinski et al., 2023; Padullés et al., 2025). The Chinese government is similarly promoting the adoption of GNSS polarimetric radio occultation payloads on the Fengyun series, thereby driving payload development and data-processing capability advances at the China Commercial Meteorological Satellite Corporation.

     Yunyao Aerospace Technology Co., Ltd., established in March 2019, is developing the "Yunyao Meteorological

Constellation" comprising 90 high–temporal-resolution meteorological satellites (Xu et al., 2025). The constellation's core payloads include GNSS radio occultation and GNSS-Reflectometry instruments. The initiative is designed to deliver an end-to-end capability—spanning spaceborne observation, data processing, and service provision—to furnish industry users with high-cadence, reliable, and scalable meteorological space-data products. The schematic in Fig. 2 illustrates the planned





network topology, highlighting on-orbit regional redundancy and enhanced temporal sampling that together support
monitoring and nowcasting of rapidly evolving meso- and small-scale weather systems.

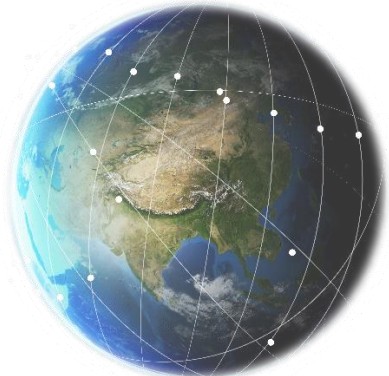

**Figure 2 The 90-satellite Yunyao Meteorological Constellation.**

On 21 March 2025, Yunyao launched its first radio-occultation satellite carrying a GNSS Polarimetric Radio
Occultation Instrument (GPROI) payload into a sun-synchronous, polar orbit (inclination 97.71°, nominal altitude ≈ 540 km).
The satellite's sub-satellite ground track on the launch date is shown in Fig. 3. From initial insertion, the spacecraft has
continuously acquired and downlinked occultation measurements, and the Yunyao Data Processing Center promptly
established the "Yunyao PRO" dataset. Following geometric and polarimetric calibration and quality control, the center
initiated a GNSS-PRO heavy-precipitation experiment and a suite of sensitivity analyses. To date, nearly three months of
observations have been used to evaluate PRO sensitivity to precipitation, including: statistical characterization of $\Delta\Phi$ profiles
across distinct weather regimes; co-consistency tests between $\Delta\Phi$ and independent ground/spaceborne references (reanalyses,
ground-based radar reflectivity, and passive-microwave retrievals); and assessment of $\Delta\Phi$ as an indicator of near-
tropospheric moisture stratification and layers of oriented frozen hydrometeors. With the feasibility and cost-effectiveness
assessments, Yunyao is considering expansion of the PRO constellation and optimization of orbital configurations to
enhance the spatiotemporal coverage and robustness of near-real-time hydro-meteorological profiling.

This paper systematically presents the technical background of the Yunyao meteorological constellation and its PRO
payload, the polarimetric radio occultation data-processing workflow and its key elements, and the sensitivity tests and
experimental results assessing PRO's response to precipitation. Section 2 describes the Yunyao occultation and polarimetric-
occultation data sources, the end-to-end processing chain and retrieval methods, and the reference products and co-
registration strategies used for cross-validation. Section 3 details the heavy-precipitation experiment design and case-
selection criteria, presents detection performance results and integrated analyses, and discusses sources of uncertainty,
methodological limitations, and potential pathways for subsequent assimilation studies.



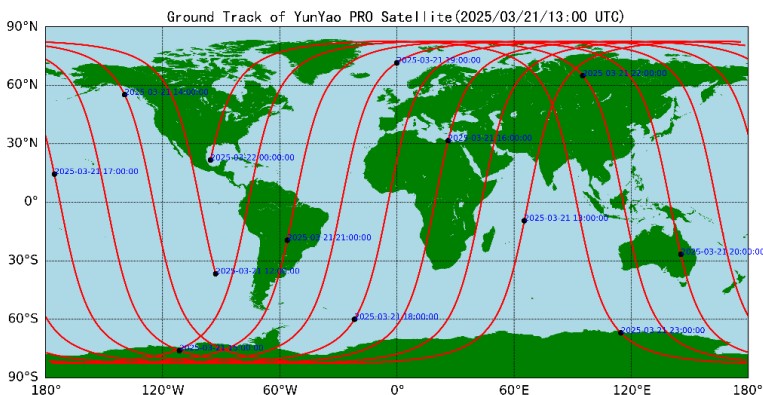

**Figure 3. The ground track of Yunyao polarimetric radio occultation satellite on 21 March 2025.**

## 2 Data and Method

### 2.1 Data collection

Between 21 March and 21 June 2025, Yunyao acquired approximately 53,000 polarimetric RO events. The global distribution of these events is shown in Fig. 4: owing to the limb-grazing geometry of GNSS radio occultation, overall spatial coverage is relatively balanced. However, parts of Eastern Europe and the Middle East exhibit pronounced data sparsity and gaps. These localized deficiencies arise not only from complex radio-frequency environments (including elevated radio frequency interference, RFI) but also from the inherently lower signal-to-noise ratio of the polarimetric RO configuration relative to conventional RO, which increases PRO's vulnerability to such adverse radio conditions. Fig. 5(a) shows the two-dimensional coverage of latitude and local time. Events exhibit a typical dual peak at dawn and dusk (approximately 04:00–06:00 and 17:00–19:00), with near symmetry across the northern and southern hemispheres. This suggests that orbital/geometric factors, rather than regional quality issues, are the primary controlling factor. Fig. 5(b) shows that the daily counts for the three major constellations are generally stable, but with a natural fluctuation of approximately 10%. The BDS averages approximately 330 daily counts, GPS approximately 230, and GLONASS approximately 110. Data reception was suspended in early April and mid-June due to a combination of constellation maintenance and receive link status. Because channel resources were not initially allocated perfectly evenly across the constellations, the actual distribution is consistent with expectations, as shown in Fig. 5(c).



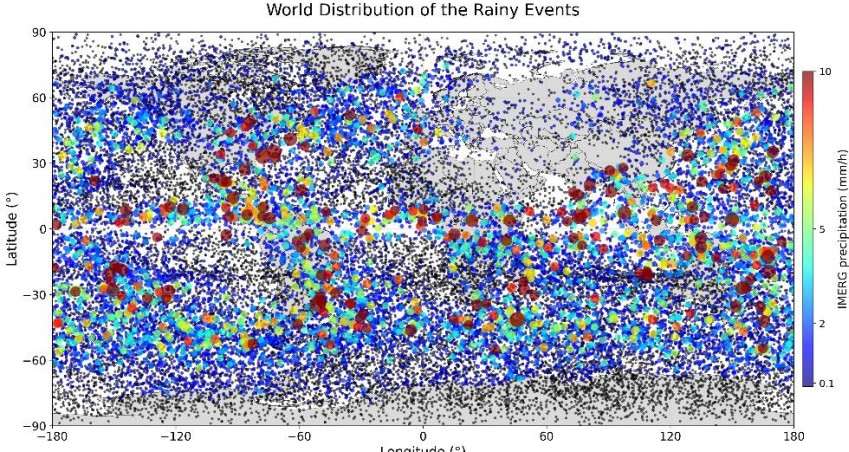

**Figure 4. Geographic distribution of Yunyao polarimetric radio occultation events from 21 March to 21 June 2025. Events are collocated with the IMERG precipitation product. Black dots denote non-precipitating events. Colored markers (blue to dark red) indicate events where IMERG precipitation > 0 mm h⁻¹; warmer colors and larger sizes indicate heavier rain.**

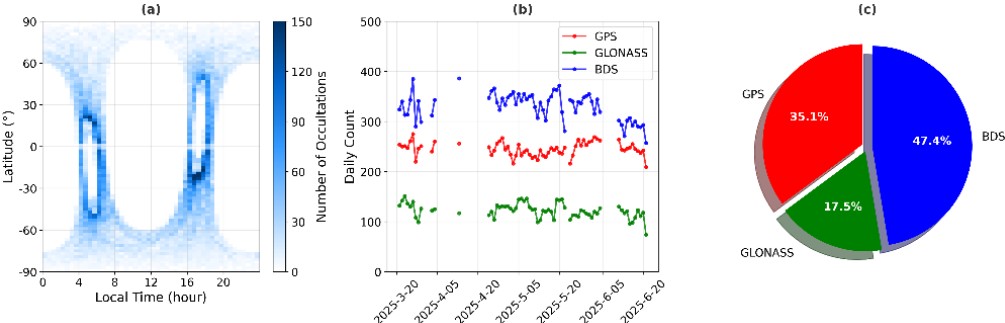

**Figure 5. (a) 1° × 30 min latitude -local time bin of PRO profiles. (b): Time series of PRO profile counts by GNSS constellation.**

## 2.2 Standard RO data-processing workflow

Fig. 6 summarizes the standard RO processing workflow at the Yunyao Data Processing Center. The workflow takes the "synthesized signal"—the complex baseband signal obtained by coherently combining the horizontal and vertical polarizations—as the scalar-channel input, sequentially deriving the excess phase, bending angle, and refractivity, and then inverting these to produce high-quality profiles of temperature, humidity, and pressure.





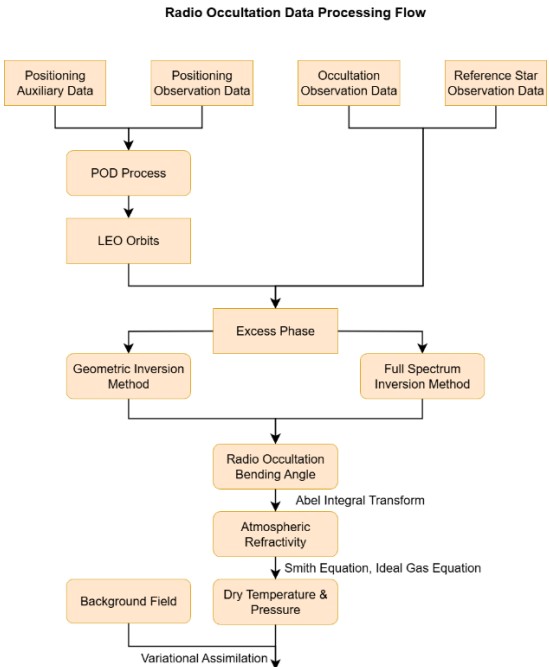

**Figure 6 Standard GNSS radio occultation processing workflow.**

## 2.2.1 Precise Orbit Determination

Precise orbit determination (POD) for low-Earth-orbit (LEO) satellites is a key driver of the accuracy of radio occultation

145  retrievals—including excess phase, bending angle, and refractivity. Accordingly, the Yunyao Data Processing Center, in collaboration with Wuhan University, devised a POD scheme tailored to Yunyao LEO meteorological satellites (Shi et al., 2008), accounting for orbital altitude, refined force models, and spacecraft characteristics such as area-to-mass ratio, volume, and solar-array design (see Table 1). Over the experimental period (21 March–21 June 2025), the orbit of POD was evaluated via the overlapping-arc method in the radial (R), transverse (T), and normal (N) components, as summarized in

150  Table 2. Because the Yunyao PRO receiver hosts both GPS and GLONASS tracking on the same receiver board, the POD accuracies for the two constellations are identical (Yue et al. 2025).

**Table 1 Precision orbit determination strategy for Yunyao LEO satellite**

| | Item | Strategies |
|---|---|---|
| | Observation | Undifferenced ionosphere-free phase and code (interval 10 s) |
| Data and products | Signal selection | GPS: L1/L2; GLONASS: L1/L2; BDS: B1/B3 |
| | Elevation cut-off | 10° |
| | GPS phase center offset | igs20.atx |



| Item | | Strategies |
|---|---|---|
| GNSS orbit and clock products | | Wuhan University ultra products (interval of 5 min) |
| Dynamic model | Arc length for orbit determination | 12 h |
| | Earth gravity field model | EIGEN_6s, 150 × 150 (Förste et at., 2010) |
| | N-body | JPL DE421 (Folkner et al., 2014) |
| | Solid Earth tides | IERS2010 (Petit et al., 2010) |
| | Earth radiation | CERES Earth radiation data (Priestley et al., 2011) |
| | Ocean tides | FES2004 (30 × 30) (Lyard et al., 2024) |
| | Atmospheric drag | DTM94 (Berger et al., 1998) |
| | Relativity | IERS2010 |
| | Solar radiation pressure coefficients | Box-Wing (Rodriguez-Solano et al., 2012) |
| Estimated parameters | LEO initial state | Position and velocity |
| | Receiver clock | Random Walk (RW) |
| | Ambiguities | Floated solution |
| | Drag coefficients | One per 1.5 h |
| | Empirical coefficients | (Piecewise periodical estimation of the sin and cos coefficients in the track and normal directions) |

**Table 2 Root Mean Square (RMS) difference for the overlap comparison in radial, transverse, normal and 3D directions for GPS GLONASS and BDS**

| GNSS system | RMS (cm) | | | |
|---|---|---|---|---|
| | R | T | N | 3D |
| BDS | 4.24 | 4.56 | 1.79 | 6.48 |
| GPS | 2.93 | 5.82 | 3.56 | 7.42 |
| GLONASS | 2.93 | 5.82 | 3.56 | 7.42 |

155 **2.2.2 Atmospheric Excess Phase Processing**

The observed GNSS carrier phase measurement on frequency can be expressed as followed:

$$L_f = \rho + c(\delta t_r - \delta t^s) + T + I_f + \lambda_f N_f + \varepsilon_f \tag{1}$$



$$\rho = \sqrt{(x^s - x_r)^2 + (y^s - y_r)^2 + (z^s - z_r)^2} \tag{2}$$

where $\rho$ denotes the geometric range between the satellite and the receiver antennas; $(x^s, y^s, z^s)$ and $(x_r, y_r, z_r)$ are the Earth-centered inertial (ECI) coordinate of the GNSS satellite and LEO receiver in the True of Date (TOD) frame, respectively; $c$ denotes the speed of light in vacuum; $\delta t_r$ and $\delta t^s$ are the receiver clock offset and satellite clock offset, respectively; $T$ denotes the neutral-atmosphere tropospheric delay; $I_f$ denotes the ionospheric delay at frequency $f$; $\lambda_f$ and $N_f$ are the carrier wavelength the integer carrier-phase ambiguity at frequency $f$, respectively; $\varepsilon_f$ contains other errors (multipath, phase center variations, etc.)

The atmospheric excess phase is then defined as

$$\Phi_{atm} = L_f - \rho - c(\delta t_r - \delta t^s) - \lambda_f N_f - \varepsilon_f = T + I_f \tag{3}$$

The single-difference phase at frequency $f$ is used by taking the phase measured to the occulting GNSS satellite and subtracting the ionosphere-free phase in Equation (4) measured from a high-elevation, non-occulting reference satellite, both tracked by the same receiver. This subtraction cancels the common receiver clock error and leaves only the differential contributions from geometric range and satellite clocks, together with the neutral-atmosphere and ionospheric delays, the integer phase ambiguity, and residual errors.

$$L_{IF}^{ref} = \frac{f_1^2}{f_1^2 - f_2^2} L_1^{ref} - \frac{f_2^2}{f_1^2 - f_2^2} L_2^{ref} \tag{4}$$

Thus, $\Delta L_f = L_f - L_f^{ref}$ preserves the atmospheric/ionospheric information along the occulting path while removing the receiver clock term, providing the starting point for excess-phase processing. We get the final excess phase for each frequency is:

$$\Phi_{atm,f} = L_f - L_{IF}^{ref} - [\Delta\rho + c\Delta\delta t^s + \lambda_f N_f - \lambda_{IF} N_{IF}^{ref} + \Delta\varepsilon_f], \tag{5}$$

where $L_f$ is the occulting-satellite carrier phase; $L_{IF}^{ref}$ is ionosphere-free combination from the reference satellite; $\Delta\rho$ and $\Delta\delta t^s$ are geometric-range and satellite-clock differences.

In the lower troposphere, GNSS receivers often switch between closed-loop and open-loop tracking to maintain lock under severe multipath. Processing coherently integrates In-phase and Quad-phase(I/Q) over defined intervals, extracts phase via arctan(Q/I) and amplitude via $\sqrt{I^2 + Q^2}$, unwraps the phase to form a continuous time series, and then merges it with closed-loop data using dedicated transition logic (Chuang et al., 2013; Sokolovskiy et al., 2006). By performing independent or combined phase reconstruction on the open-loop observations I/Q of vertical and horizontal polarization signals, we can obtain the vertical-/horizontal-polarization excess phases or the combined ones. The former is used to obtain polarization phase differences or polarization observation occultation profiles based on the traditional RO retrieval algorithm, and the latter is processed similarly to standard RO.



### 2.2.3 Atmospheric Profile Inversion

Our atmospheric-profile inversion adopts a dual-method strategy tailored to altitude. Above 20 km, bending angles are retrieved with geometric optics (GO) from Doppler-derived excess phase; below 20 km, full-spectrum inversion (FSI)

transforms the complex RO signal into impact-parameter space (Gorbunov et al., 2004; Gorbunov et al. 2006), resolves multipath, and outputs Local Spectral Width (LSW) sequences used for quality control.

Ionospheric correction is altitude dependent. For regions above 20 km, the first-order ionosphere is removed with the ionosphere-free combination.

$$\alpha_{corrected}(a) = \alpha_{L1}(a) + \frac{f_1^2}{f_1^2 - f_2^2}(\alpha_{L1}(a) - \alpha_{L2}(a)), \tag{6}$$

where $\alpha_{L1}(a)$ and $\alpha_{L2}(a)$ are bending angles derived from L1 and L2 frequencies. For regions above 20 km, we employ a fixed correction application strategy.

$$\alpha_{corrected}(a) = \alpha_{L1}(a) + \Delta\alpha_{corrected}(a = a_{20km}), \tag{7}$$

where $\Delta\alpha_{corrected}(a = a_{20km})$ is the ionospheric correction at the 20 km impact parameter.

To obtain a physically consistent bending-angle profile, we merge GO- and FSI-retrieved bending angles the geometric

optics (GO) method (upper/middle atmosphere) based on Doppler shift analysis and the Full Spectrum Inversion (FSI) method (lower atmosphere) with a smooth window around 20 km. The weight formula is:

$$\alpha_{combined}(a) = W(a)\alpha_{GO}(a) + (1 - W(a))\alpha_{FSI}(a), \tag{8}$$

where $a$ is the impact parameter; $W(a)$ is a smooth window increasing from 0 to 1 near ~20 km; $\alpha_{GO}(a)$, $\alpha_{FSI}(a)$ are the GO- and FSI-retrieved bending angles; $\alpha_{combined}(a)$ is the fused bending.

LSW from FSI is used for threshold-based quality control to flag multipath. The merged profile is statistically optimized against a background using covariance weighting.

$$\alpha_{opt} = \alpha_{obs} + C_{obs}(C_{obs} + C_{bg})^{-1}(\alpha_{bg} - \alpha_{obs}), \tag{9}$$

where $\alpha_{obs}$ is the observation; $\alpha_{bg}$ is the background bending; $C_{obs}, C_{bg}$ are the corresponding error covariances; $\alpha_{opt}$ is the optimized bending used for inversion.

The optimized bending angle is converted into refractive index by top-down integration of the inverse Abel transformation corrected for Earth ellipticity. The optimized bending angles are inverted to refractivity via the Abel transform with top-down integration and Earth-ellipticity correction:

$$N(x) = \frac{1}{\pi}\int_x^\infty \frac{\alpha(a)}{\sqrt{a^2 - r^2}}da, \; x = nr, \tag{11}$$

where $N(x)$ is refractivity multiplied by radius; $n$ is refractive index; $r$ is geocentric radius. Dry temperature and pressure

profiles are derived from refractivity using the Smith-Weintraub relation in combination with the hydrostatic balance and the ideal gas law, integrated downward from a prescribed top-of-atmosphere boundary condition. To obtain full profiles of temperature, pressure, and humidity, we apply a one-dimensional variational (1D-Var) retrieval, in which GNSS-RO refractivity serves as the observation constraint and the Yunyao Meteorological Numerical Forecast Model provides the



background state. The minimum-variance (optimal-estimation) solution yields vertically resolved fields of temperature,
pressure, and humidity.

## 2.3 PRO data-processing workflow

YunYao in-house GNSS-RO receiver onboard the satellite continuously acquires RO data at 100 Hz. In the YunYao
configuration, only the occulting link is tracked, with closed-loop and open-loop channels running in parallel to allow
flexible handover control in post-processing. Each RO event is tracked independently by two polarization ports (H and V),
which operate independently yet are time-synchronized and output in parallel. The processing strategy and product
convention for PRO follow PAZ (Padullés et al., 2024), and the overall workflow is illustrated in Fig. 7.

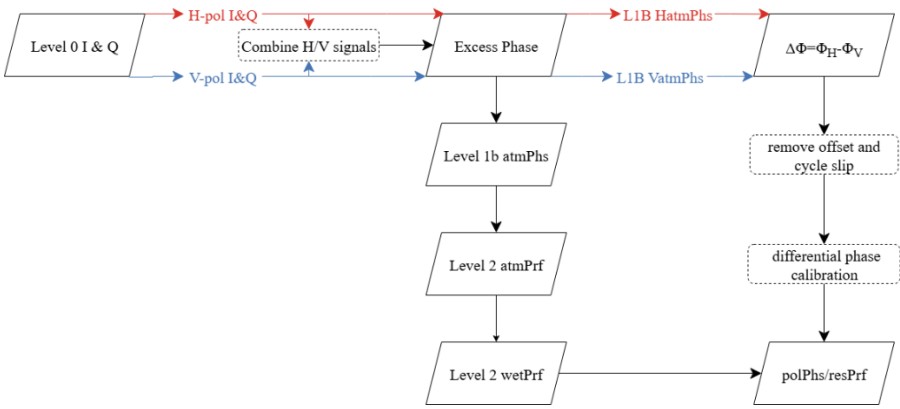

**Figure 7. GNSS Polarimetric Radio Occultation Processing Workflow.**

The processing chain comprises four stages: (i) I/Q synchronization and geometric co-registration;(ii) generation of
standard RO L1B products;(iii) construction and calibration of the polarimetric differential phase;(iv) cycle-slip removal,
smoothing, and detrending to produce L2 products.

The H- and V-channel I/Q streams are time- and frequency-aligned and coherently combined into a complex baseband;
signal-to-noise ratio (SNR) is computed and harmonized, and epochs and sampling rates are unified to ensure strict
consistency for differencing and parallel processing. Standard RO processing is then applied independently to the H, V, and
"synthesized" branches to produce the L1B atmospheric excess phases HatmPhs, VatmPhs, and atmPhs, as well as the
polarimetric differential phase defined as the $\Delta\Phi$ between the atmospheric excess phases from the two polarization ports:

$$\Delta\phi = \phi_H - \phi_V, \tag{12}$$

This includes single differencing against a reference link to remove receiver clock and hardware delays, the dual-frequency
ionosphere-free combination to suppress first-order ionospheric effects, and detection/repair of carrier cycle slips with
lock-loss detection and phase reconstruction.



The PRO observable is defined as the $\Delta\Phi$ between the atmospheric excess phases from the two polarization ports:

In the absence of polarizing media along the path, $\Delta\Phi$ should remain constant in time. Under ideal RHCP illumination the absolute H–V phase offset is $\pi/2$; however, because H and V are tracked independently, only a random constant remains in the measurements. We estimate this constant over a stable stratospheric segment and remove it to align the zero level of

$\Delta\Phi$. Even after conventional per-port cycle-slip repair, forming $\Delta\Phi$ can expose residual jumps. We therefore adopt the Padullés CL/OL de-slipping scheme (Padullés et al., 2020; Ao et al., 2009): half-cycle slips are corrected during CL and full-cycle slips during OL. The Formulas follow:

$$\Delta\Phi(t) = \arctan\left(\tan\left(\Delta\Phi(t)\right)\right) \tag{13}$$

$$\Delta\Phi(t) = \text{arctan2}\left(\sin\left(\Delta\Phi(t)\right), \cos\left(\Delta\Phi(t)\right)\right). \tag{14}$$

This procedure removes remaining half- and full-cycle slips. Fig. 8 shows an example for a GPS event (PRN-31) on 2025-04-29: panel (a) presents calibrated L1 SNR for H and V, decreasing as the tangent point descends and showing stronger jitter at the CL–OL transition (dashed lines); panel (b) compares the raw and de-slipped $\Delta\Phi$, where the corrected series becomes continuous and nearly constant aloft; panel (c) plots the corrected $\Delta\Phi$ converted to path length together with a 1-s moving average.

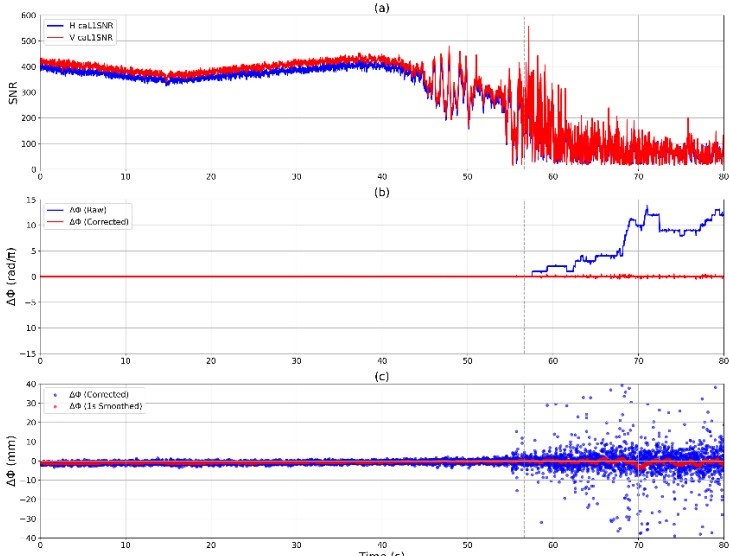


**Figure 8. (a) L1 signal-to-noise ratio time series for the H- and V-polarization channels; (b) time series of the raw differential phase $\Delta\Phi$ (blue) and the cycle-slip–corrected $\Delta\Phi$ (red); (c) time series of the corrected $\Delta\Phi$ (blue) and its 1 s moving-average smoothed series (1 s Smoothed). Gray dashed lines indicate the closed-loop–open-loop transition in each panel.**

Afterwards, $\Delta\Phi$ is smoothed with a 1-s moving window to suppress high-frequency noise and linearly detrended along

the full profile to produce the polPhs file (Padullés et al., 2024). In parallel, the excess phase from the synthesized signal undergoes standard RO retrieval to derive dry and wet profiles, which are interpolated to a 0.1-km grid to form the resPrf file (Padullés et al., 2024) for collocation of $\Delta\Phi$ with thermodynamic fields. For calibration, IMERG-identified rain-free events are used to derive the in-orbit antenna pattern and to remove any residual ionospheric imprint on $\Delta\Phi$ (Padullés et al., 2020).




Because the ΔΦ originates from hydrometeors (rain, cloud, ice crystals), we set ΔΦ to zero at 30 km under water-free
conditions to remove a profile-wide offset; all subsequent ΔΦ values are referenced to this level. The calibrated phase is
again smoothed with a 1-s filter and linearly detrended; the resulting calibrated and smoothed ΔΦ is the primary PRO
observable. After these steps, remaining ΔΦ variability can be attributed to differences in H- and V-component propagation
induced by non-spherical, preferentially oriented hydrometeors along the path. Fig. 9 illustrates the smoothing and
detrending: in (a) the light-blue curve is ΔΦ after de-slipping, the blue curve is the 1-s smoothed ΔΦ, and the red dashed line
is the linear trend; in (b) the detrended ΔΦ after smoothing is shown.

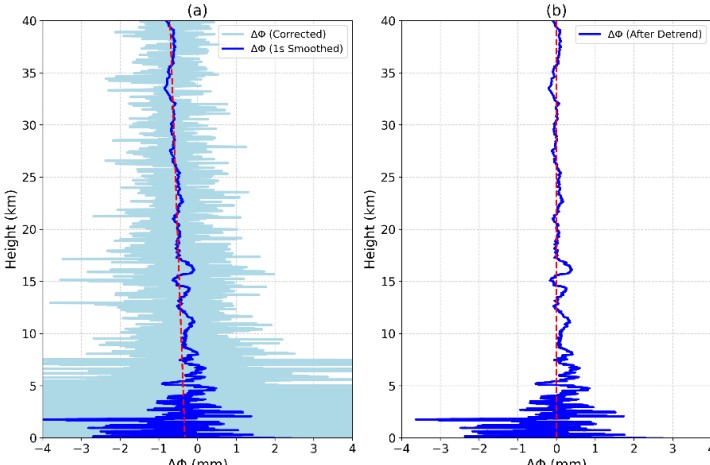

**Figure 9. Example of linear trend correction applied to the differential phase.**

## 2.4 GPM

In the calibration and validation of PRO, the correlation between polarimetric occultation data and precipitation products
provides an independent assessment of rainfall influence. Rain-free events serve as references to calibrate systematic biases,
while rainy events test ΔΦ sensitivity. IMERG, which fuses multi-satellite passive microwave and infrared observations,
offers reliable, global precipitation estimates at ~0.1° × 0.1° spatial and 30-min temporal resolution, with both near-real-time
and final products (Katona et al., 2025; Watters et al., 2024). Its coverage and resolution make it suitable for independent co-
location and sample selection.
Unlike prior approaches that fixed circular regions (Katona et al., 2025), we project the actual occultation ray paths—
tilted elongated bands due to GNSS–LEO relative motion—onto the surface, considering only the troposphere below 6 km
where precipitation primarily affects propagation. Polygonal footprints thus reflect both along-ray elongation and lateral
extent, reducing geometric misclassification. Each occultation event is temporally matched to the nearest IMERG 30-min
interval, and the surface-projected ray points are weighted to obtain an event-level mean precipitation. Rain-free events are
used for calibration of the differential phase baseline, while rainy events assess PRO sensitivity and discriminative





capability. Although only the lower-troposphere projection is used, the long-term dataset provides sufficient statistics for robust validation.

**2.5 ERA5**

ERA5 is the fifth-generation global atmospheric reanalysis dataset developed and maintained by the European Centre for
Medium-Range Weather Forecasts (ECMWF), with a horizontal resolution of approximately 30 km and 137 vertical levels spanning the surface to 80 km. Covering the period from 1979 to the present, ERA5 undergoes rigorous quality control, with near-real-time products released within three months and reanalysis updates available with a six-day lag. ERA5 data can be accessed and downloaded via the Copernicus Climate Data Store (https://cds.climate.copernicus.eu/cdsapp#!/home) (Hersbach et al. 2020). For this study, the dataset corresponding to the temporal span of the polarimetric occultation
observations (21 March–22 June 2025) was retrieved and stored on the servers of Yunyao in netCDF format.

**3. The correlation experiment of PRO profile and precipitation**

**3.1 Quality evaluation of PRO profile**

H-polarized, V-polarized, and their combined circularly polarized observation data can all be used to retrieve meteorological elements, and the retrieved profiles are nearly identical. Compared to H- and V-polarized observations, the synthesized data
exhibit a higher signal-to-noise ratio and a greater success rate in retrieval. Therefore, the accuracy of the profiles derived from the synthesized observations is representative of the performance of polarimetric occultation. Using ERA5 as a reference, a comparative evaluation was performed on the profiles of bending angle, refractivity, temperature and pressure retrieved from the polarimetric occultation.

The formula for calculating refractivity is given:

$$N = k_1 \times \frac{P-e}{T} + k_2 \times \frac{e}{T} + k_3 \times \frac{e}{T^2}, \tag{15}$$

where $N$ is the refractivity (in N-units), $P$ is pressure (in hPa), $T$ is the temperature (in K), and $e$ is the water vapor pressure (in hPa). The coefficients are $k_1$ = 77.604 K hPa$^{-1}$, $k_2$ = 64.79 K hPa$^{-1}$, and $k_3$ = 377600.0 K$^2$ hPa$^{-1}$.

The profile data corresponding to PRO profile is calculated by applying logarithmic cubic spline interpolation in the vertical dimension and bilinear interpolation in the horizontal plane to the ERA5 data. The relative deviations of the retrieved
profiles (including bending angle, refractivity, temperature, and pressure) and the ERA5 values are then computed using Equation (10), as shown below.

$$R = \frac{O-B}{B} \times 100\%, \tag{16}$$

where $R$ represents the relative deviation, $O$ denotes the observed value, and $B$ represents the reference value.





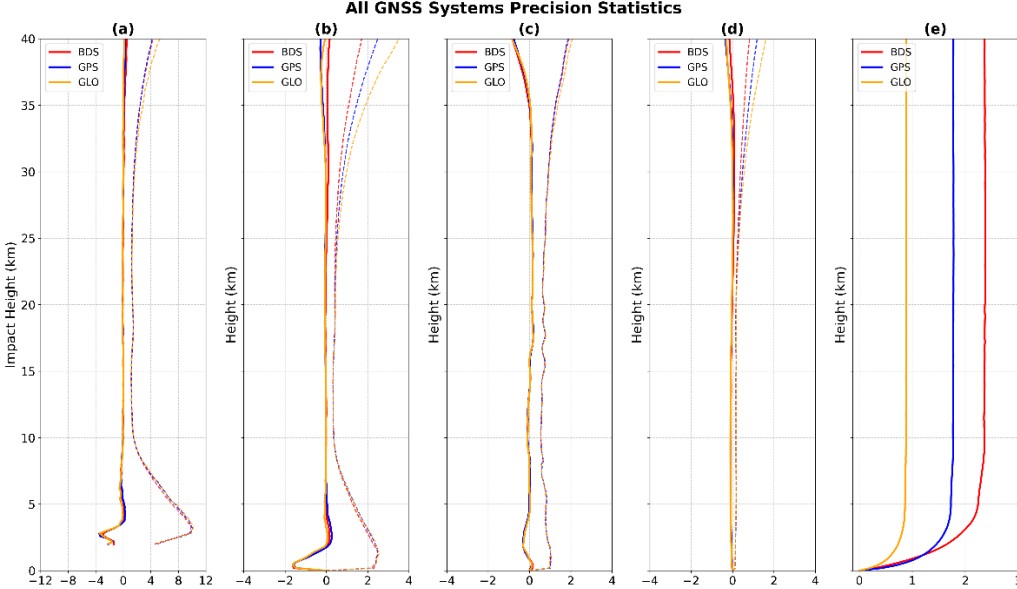

**Figure 10. Comparison between PRO profile and ERA5 data.**

Fig. 10 shows the accuracy of PRO profile between March 21 and June 21, 2025. Statistical analyses were performed on the deviations of bending angle, refractivity, temperature, and pressure profiles at various altitudes for the BDS, GPS, and GLONASS systems. Overall, all three systems exhibit minimal bias and stable errors from the mid-upper troposphere to the lower stratosphere (approximately 2–30 km). In contrast, the uncertainty increases significantly in the boundary layer below 1 km and the upper stratosphere above 30 km, which is generally consistent with the characteristic error profile of conventional radio occultation retrievals.

The accuracy of bending angle and refractivity profiles is shown in Fig. 10(a) and 10(b), respectively. Both exhibit similar error variation characteristics. Taking the impact height as the y-axis in Fig. 10(a), the bending angle accuracy shows its most pronounced negative bias at 2–4 km, where the mean reaches about −2.9% and the standard deviation peaks near 9.6%, consistent with multipath and super-refraction. From 4 to 30 km the bias is slightly positive and generally decreases with height; above 30 km both the mean and the spread increase modestly and the curves begin to diverge. In Fig. 10(b), which uses altitude on the y-axis, refractivity exhibits a modest positive near-surface bias and large variability due to sharp boundary-layer moisture gradients and residual multipath; between 2 and 10 km, absolute mean differences are typically below 0.5% and decrease with height; from 10 to 30 km, means remain within 0–0.2% with a spread of about 1%, followed by slight increases and divergence above 30 km.

Figure 10(c) shows the temperature profile accuracy. A slight negative bias appears between 2-3 km, transitions to a positive bias at 10–18 km, and stabilizes near zero from 20-30 km. This pattern reflects the propagation of multipath and humidity-induced errors from the lower troposphere into the retrieval. Above 30 km, the negative bias is dominated by higher-order ionospheric residuals. Figure 10(d) presents the pressure profile accuracy. A small negative bias persists



throughout the column, reaching about −0.07% near the tropopause. The bias increases slightly between 30–40 km due to error amplification from sequential integration, but remains small (2-30 km).

In summary, the thermodynamic elements retrieved from the synthesized PRO data demonstrate robust performance between 2–30 km, with both bias and dispersion remaining small.

## 3.2 Rainfall detection capability of PRO profile

This section presents the differential phase observed by the PRO payload in relation to precipitation. The detection capabilities are similar across different GNSS constellations.

To evaluate the precipitation detection capability of the Yunyao PRO profile, we matched the $\Delta\Phi$ along the ray path with the IMERG precipitation rate product from GPM. The geometric relationship was constrained using the tangent point trajectory in the longitude-latitude plane. Fig. 10 and Fig. 11 present two representative cases, illustrating scenarios "$\Delta\Phi$

without precipitation" and "$\Delta\Phi$ with precipitation", respectively, to verify the correlation between precipitation and $\Delta\Phi$.

In the non-precipitation case, Fig. 11(a) shows that the ray path remains entirely within the IMERG zero-precipitation region. The corresponding vertical profile of $\Delta\Phi$ in Fig. 11(b) exhibits minor fluctuations around zero across all altitude layers, with mean $\Delta\Phi$ values of −0.05 mm and −0.09 mm for the 0–6 km and 0–12 km ranges, respectively. These magnitudes are consistent with the slight negative bias observed in the profile and can be attributed to noise and residual

correction errors rather than hydrometeor signals. Under the same processing chain and geometric conditions, the sub-millimeter mean $\Delta\Phi$ can be regarded as the noise background of the instrument and retrieval system. There is no significant systematic deviation, indicating that the calibration is accurate.

In contrast, as shown in Fig. 12(a), the ray path traverses multiple intense precipitation regions starting from the near-surface layer. Fig. 12(b) reveals a significant increase in $\Delta\Phi$ within the troposphere, with rapid attenuation observed near the

ground. A peak value of 15–16 mm occurs around 3–4 km, decreasing to approximately 6–8 mm at 6 km, further declining to 1–2 mm between 8–10 km, and approaching zero at 14–18 km. The mean $\Delta\Phi$ values over the 0–6 km and 0–12 km layers are 12.25 mm and 6.96 mm, respectively, with the matched IMERG precipitation rates being 2.15 mm h⁻¹ and 1.13 mm h⁻¹. This vertical structure, which is characterized by strong signals in the lower atmosphere and rapid attenuation with height, is consistent with the distribution of raindrop-dominated liquid water and its geometric anisotropy: large, oblate, and oriented

raindrops in the near-surface layer cause significant $\Delta\Phi$ accumulation at L-band, while the liquid water content decreases rapidly with altitude and the contribution from ice-phase particles remains weak, leading to $\Delta\Phi$ approaching zero. The correlation between precipitation and $\Delta\Phi$ is also confirmed: tangent points at lower altitudes are closer to the main precipitation core, corresponding to higher $\Delta\Phi$ peaks; as the tangent points move away from the precipitation core with increasing height, $\Delta\Phi$ decreases accordingly.

It should be noted that $\Delta\Phi$ is an integrated quantity along the oblique path, influenced by factors such as the horizontal drift of tangent points and the inhomogeneity of the precipitation volume. The spatiotemporal representativeness error of IMERG may also affect the matched precipitation rate. The consistency in geometry, vertical structure, and external precipitation



fields between these two cases clearly demonstrates that the Yunyao PRO profile exhibits a distinct and reproducible response characteristic to liquid precipitation.

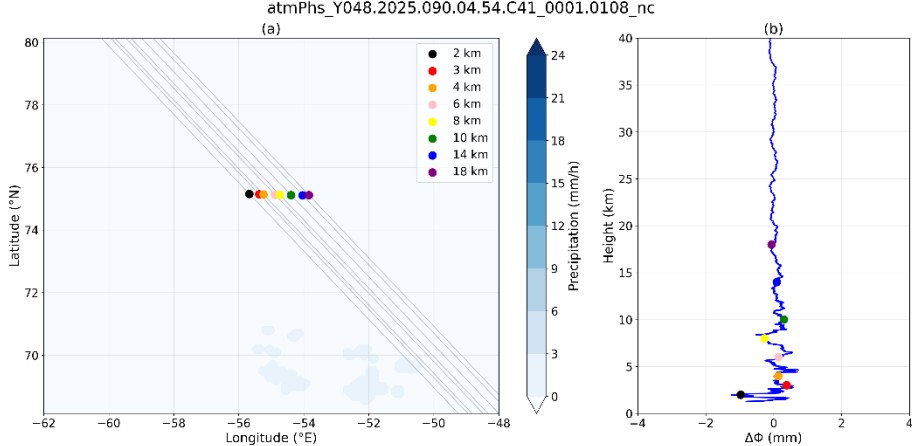

Figure 11. ΔΦ without precipitation-induced

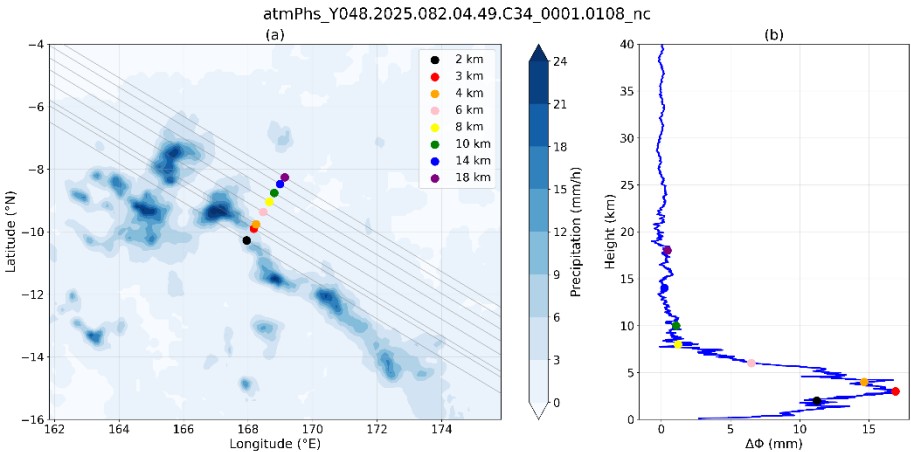

Figure 12. ΔΦ with precipitation-induced

## 3.3 Precipitation sensitivity experiment of PRO profile

Based on the registration statistics between ΔΦ of Yunyao PRO profile and precipitation of GPM IMERG, only the segment below 6 km altitude was used. The ΔΦ profiles smoothed with a 1-second window were interpolated onto a 100-meter equidistant vertical grid from 0 to 40 km. The profiles were then grouped according to precipitation intensity: no precipitation (0 mm h⁻¹), light precipitation (0–0.1, 0.1–0.2, 0.2–0.3, 0.3–0.5, 0.5–0.7 mm h⁻¹), and moderate-to-heavy precipitation (0.7–1, 1–2, 2–3, 3–5, 5–7, ≥7 mm h⁻¹). For each group, the mean and standard deviation of ΔΦ were computed at each altitude layer. Fig. 13(a)–(c) show the sample size, mean ΔΦ, and the standard deviation profile under non-precipitating conditions for each precipitation interval. This method follows previous studies (Teunissen et al., 2017;



Cardellach et al., 2018; Gorbunov et al., 2011), enabling direct comparison with published results.

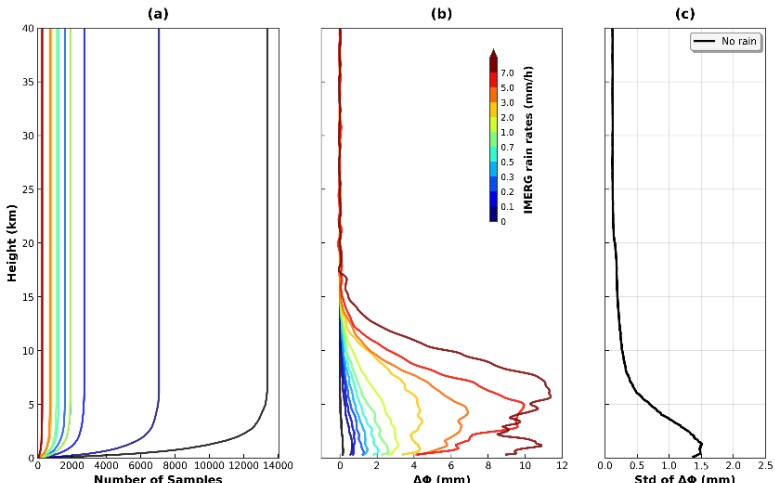

**Figure 13. Statistical analysis of results over 0–40 km. (a) Sample size for each precipitation group; (b) Mean ΔΦ per precipitation group, with the black curve indicating the bias under non-precipitating conditions; (c) Standard deviation of the ΔΦ for the non-precipitation (0 mm h⁻¹) group.**

Fig. 13(a) shows the distribution of sample sizes across altitude and precipitation intensity groups. The non-precipitation group contains approximately 13,000 samples per layer between 6–15 km, while several light precipitation 390 groups (0–0.5 mm h⁻¹) maintain sample sizes on the order of several thousand. In contrast, heavier precipitation groups (2–7 mm h⁻¹) exhibit an order-of-magnitude decrease in sample count. This sample size structure reflects both the spatiotemporal sparsity of global precipitation and ensures the robustness of mean and variance estimates for each group.

Fig. 13(b) presents the mean ΔΦ profiles for different precipitation intensity intervals. The non-precipitation group shows ΔΦ values close to 0 mm across all altitude layers, as expected, and is consistent with the results reported by 395 Cardellach et al. (2014). As precipitation intensifies, the magnitude of ΔΦ increases and exhibits significant altitude dependence. At a representative altitude of 3 km, the mean ΔΦ values for each group—from light to heavier precipitation—are 0.07, 0.48, 0.89, 1.23, 1.54, 2.07, 2.81, 4.27, 6.69, 9.54, and 9.03 mm, respectively. These results indicate an approximately linear response of ΔΦ to path-averaged precipitation rate, with higher sensitivity in the lower troposphere. It should be noted that the ΔΦ profiles of the heavier precipitation groups peak around 3–5 km and then decrease rapidly with 400 height, falling below 1–2 mm above 10–12 km. This vertical structure which characterized by strong signals at lower levels and rapid attenuation aloft reflects both the optical anisotropy dominated by liquid raindrops and geometric factors: when the occultation tangent point is near 5–6 km, the ray path achieves a longer effective propagation distance within a relatively homogeneous cloud and precipitation region, thereby maximizing the integrated contribution to ΔΦ. Such behavior of the ΔΦ reveals the coupling between strong liquid-phase contributions in the lower atmosphere and weak ice-phase 405 contributions at higher altitudes, and is consistent with the geometric mechanism of "ray path length maximization."

Fig. 13(c) shows the altitude-dependent standard deviation of ΔΦ under the 0 mm h⁻¹ condition (non-precipitation



regions), which can be regarded as the "noise background" of the instrument and retrieval system. This provides a statistical basis for batch quality control and threshold setting. The standard deviation decreases with increasing altitude: it is approximately 1.5 mm near the surface,the result generally consistent with Padullés et al. (2020), though the near-surface
trend differs, likely due to factors such as the Yunyao PRO receiver performance or the open-loop tracking algorithm. The value decreases to around 1.40 mm at 2 km, 0.70 mm at 5 km, and remains below 0.30 mm above 10 km, stabilizing between 0.12–0.16 mm in the 20–30 km range. This magnitude is consistent with the results reported by Padullés et al. (2020), further validating the stability of calibration and correction.

It should be noted that in this study, the precipitation-based grouping using the IMERG product was performed based
on the path-averaged precipitation rate below 6 km altitude. Therefore, the correspondence between $\Delta\Phi$ and precipitation reflects a coupling of path-integrated and area-averaged relationships. Inhomogeneity in the precipitation structure and beam-filling effects may introduce a certain degree of dispersion, particularly in the lower atmosphere. Nevertheless, the consistency in geometry, vertical structure, and magnitude of the grouped profiles along with their good agreement with previous literature collectively demonstrates that the $\Delta\Phi$ of Yunyao PRO profile exhibits a distinct, reproducible, and
dynamically responsive detection capability for liquid precipitation.

To further quantify the detection performance of $\Delta\Phi$ in 0–6km, we computed

$$P(\Delta\Phi > \tau \,|\, R), \tag{17}$$

where $R$ is the path-averaged precipitation rate from IMERG, and $\tau$ denotes the phase difference threshold (0.5, 1, 2, and 5 mm). As shown in Table 3, the threshold exceedance rate increases monotonically with R. Under non-precipitating
conditions, only 7.85% and 2.25% of the cases exceed 1 mm and 2 mm, respectively, while exceedance of 5 mm is nearly negligible (0.09%). However, approximately 17% of cases still show $\Delta\Phi$ exceeding 0.5 mm, indicating that additional factors influencing $\Delta\Phi$ correction need to be considered. Under light precipitation, the exceedance ratio rises. When $R \geq 1$ mm h$^{-1}$, the rates exceeding 1 mm and 2 mm reach 75.40% and 52.44%, respectively. For $R = 2$–5 mm h$^{-1}$, these values increase further to 89.37% and 76.29%. When $R > 5$ mm h$^{-1}$, the corresponding rates are 97.91% and 95.20%, with 77.61%
of cases exceeding 5 mm. Therefore, a threshold of 1 mm can be used as a highly sensitive indicator for "precipitation presence", 2 mm as a conservative and highly specific threshold for "confident detection", and 5 mm as an indicator for heavy precipitation/deep convection.

Precipitation is assessed using inverse conditional probability:

$$P(R > \rho \,|\, \Delta\Phi). \tag{18}$$

As shown in Table 4, when $\Delta\Phi < 0.1$ mm, the probability of R > 0.1 mm h$^{-1}$ is only 1.85%, and R > 1 mm h$^{-1}$ is nearly zero. This indicates that small phase differences are highly indicative of non-precipitation scenarios. For $\Delta\Phi$ in the range of 0.1–1 mm, the probability of R falling within 0.1–1 mm h$^{-1}$ is 61.71% (with 16.34% for the sub-interval), demonstrating moderate predictive capability. When $\Delta\Phi = 1$–2 mm, the probability for R = 1–2 mm h$^{-1}$ increases to 72.17% (41.36%). For $\Delta\Phi = 2$–5 mm, the probabilities of R > 1 mm h$^{-1}$ and R > 2 mm h$^{-1}$ reach 87.90% and 74.53%, respectively. When $\Delta\Phi > 5$ mm, the





probabilities for R > 1 mm h⁻¹, R > 2 mm h⁻¹, and R > 5 mm h⁻¹ are 97.18%, 94.34%, and 75.63%, respectively, indicating a high likelihood of significant heavy precipitation.

**Table 3. Statistics on some ΔΦ relative thresholds for each elevation layer**

| rain rates range (mm h⁻¹) | The ratio of exceeding ΔΦ (%) | | | |
|---|---|---|---|---|
| | 0.5 | 1 | 2 | 5 |
| 0 | 16.98 | 7.85 | 2.25 | 0.09 |
| 0-0.1 | 37.97 | 21.44 | 7.81 | 0.51 |
| 0.1-1 | 50.10 | 31.84 | 13.07 | 1.02 |
| 1-2 | 86.19 | 75.40 | 52.44 | 9.71 |
| 2-5 | 93.95 | 89.37 | 76.29 | 29.73 |
| >5 | 99.07 | 97.91 | 95.20 | 77.61 |

**Table 4. Statistics on some rain rates relative thresholds for each elevation layer**

| ΔΦ range (mm) | The ratio of exceeding rain rates (%) | | | | |
|---|---|---|---|---|---|
| | 0.01 | 0.1 | 1 | 2 | 5 |
| 0.1 | 9.18 | 1.85 | 0.04 | 0.00 | 0.00 |
| 0.1-1 | 74.94 | 61.71 | 16.34 | 5.65 | 0.73 |
| 1-2 | 94.55 | 91.18 | 72.17 | 41.36 | 8.15 |
| 2-5 | 97.43 | 96.14 | 87.90 | 74.53 | 20.00 |
| >5 | 99.70 | 99.38 | 97.18 | 94.34 | 75.63 |

Fig. 14 shows boxplots of precipitation intensity versus ΔΦ, with the path-averaged precipitation rate *R* along the occultation ray path as the independent variable. The mean and variability of the layer-averaged ΔΦ over 0–6 km and 0–10 km are presented as functions of *R*. Both exhibit pronounced monotonic and convex growth curves. Taking the 0–6 km precipitation intensity–phase difference relationship as an example, ΔΦ fluctuates slightly around zero under near-zero precipitation conditions. When *R* = 0.1–0.3 mm h⁻¹, ΔΦ increases to approximately 0.3–0.8 mm. Beyond *R* ≥ 1 mm h⁻¹, the curve rises sharply, with *R* ≈ 3, 5, 7, and 10mm h⁻¹ corresponding to ΔΦ values of approximately 3, 5, 7, and 9 mm, respectively. The expansion of error bars with increasing R reflects stronger spatial inhomogeneity in heavy precipitation and the variability introduced by geometric path weighting.

The difference between the 0–6 km and 0–10 km statistical results reflects the influence of altitude on the correlation. The ΔΦ values in the 0–10 km is systematically lower than those in the 0–6 km panel at the same *R*, indicating that the inclusion of higher altitudes (>6 km) "dilutes" the lower-layer polarimetric signal dominated by liquid precipitation, this is a direct consequence of path-integration. This result is consistent with the physical picture established in previous literature, where liquid water dominates and lower-layer contributions prevail: the shape/orientation anisotropy of raindrops in the near-surface layer, combined with longer cloud-path lengths, amplifies the ΔΦ accumulation. In contrast, the decrease in





liquid water content and the weaker contribution from ice-phase particles at higher altitudes result in a continued yet diminishing marginal increase in the curve.

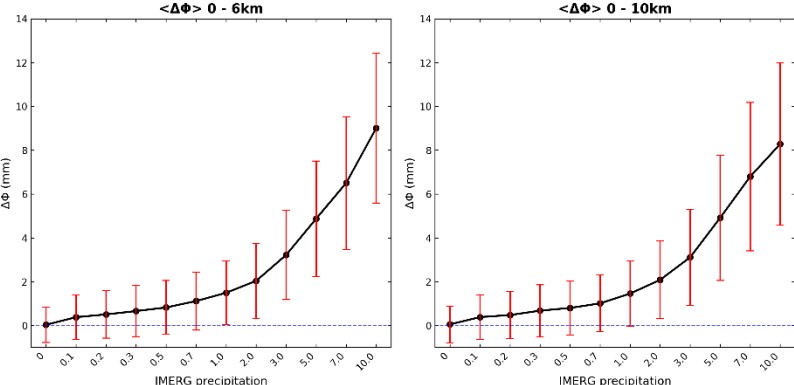

**Figure 14. The IMERG precipitation intensity-phase difference statistics of the 0-6km and 0-10km datasets.**

Meanwhile, we also performed binning of $\Delta\Phi$ based on the 0–6 km layer average (e.g., 0–0.1 mm, 0.5–0.7 mm, …, ≥18 mm). Within each $\Delta\Phi$ bin, only samples whose R values fell within the range of mean ± 1σ were retained for quality control, thereby obtaining a conditional mapping of $\bar{R}$ ($\Delta\Phi$) ± σ(R). This mapping provides the most probable precipitation rate and its uncertainty when a specific $\Delta\Phi$ is observed, as shown in Table 5.

**Table 5. The relationship of Polarization phase difference and rain rate**

| PRO $\Delta\Phi$(mm) | mean(mm) | std(mm) | PRO $\Delta\Phi$(mm) | mean(mm) | std(mm) |
|---|---|---|---|---|---|
| 0-0.1 | 0.03 | 0.05 | 7-8 | 2.94 | 1.03 |
| 0.1-0.2 | 0.04 | 0.06 | 8-9 | 3.12 | 1.03 |
| 0.2-0.3 | 0.04 | 0.06 | 9-10 | 4.19 | 1.12 |
| 0.3-0.5 | 0.05 | 0.10 | 10-11 | 4.26 | 1.14 |
| 0.5-0.7 | 0.07 | 0.15 | 11-12 | 4.51 | 0.87 |
| 0.7-1 | 0.10 | 0.21 | 12-13 | 4.83 | 0.98 |
| 1-2 | 0.16 | 0.24 | 13-14 | 4.99 | 1.00 |
| 2-3 | 0.20 | 0.56 | 14-15 | 4.66 | 1.28 |
| 3-4 | 0.48 | 0.83 | 15-16 | 7.28 | 0.96 |
| 4-5 | 0.78 | 0.88 | 16-17 | 6.87 | 1.65 |
| 5-6 | 0.93 | 1.33 | 17-18 | 7.67 | 0.74 |
| 6-7 | 2.40 | 1.15 | >18 | 8 | 0.6 |

## 4 Conclusions

This study establishes the first end-to-end framework for PRO observations from the Yunyao mission. Quality assessment confirms that the synthesized H-V signals achieve low noise and bias in bending angle and refractivity within 2-30 km, with stable temperature and pressure retrievals; boundary-layer and upper-stratospheric error patterns are consistent with previous findings, demonstrating that the conventional RO capability meets operational standards and provides a reliable baseline for



interpreting $\Delta\Phi$. For precipitation sensitivity, we introduce a collocation method based on realistic ray geometry, in which only the ray segments below 6 km are matched with IMERG using temporal and spatial weighting, effectively reducing the geometric mismatches of fixed circular windows. Case studies further reveal a distinct $\Delta\Phi$ vertical structure: near-zero under

rain-free conditions, but peaking at several to over ten millimeters at 3-5 km within precipitation, followed by rapid attenuation to 1-2 mm above 10-12 km. These results, together with statistical analyses, support the use of $\Delta\Phi$ as a robust precipitation-sensitive observable and motivate further development of forward operators, assimilation schemes, and constellation-based applications.

*Author contributions*. All authors contributed to the conceptualization and design of the experiments, data processing, and analysis. L.K. led the data processing and analysis. F.H.L. contributed to conceptualisation, methodology and interpretation. N.F.F., Y.C., S.X., and B.B.X. contributed to data analysis, validation of the data and writing.

*Competing interests*. The authors declare no competing interests.

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
