# Peer review of "On-Orbit Calibration and Performance Validation of the Yunyao Polarimetric Radio Occultation System"

_EGUsphere, 2025_

## Referee Comment (RC1)

**On-Orbit Callibration and Performance Validation of the Yunyao Polarimetric Radio Occultation System**

**General comments**

The manuscript presents the first on-orbit calibration, validation, and performance assessment of the Yunyao Polarimetric Radio Occultation (PRO) system. The study describes the data processing methodology, analyzing over 53000 occultation events collected between March and June 2025, and employs the GPM IMERG precipitation product to establish an empirical relationship between the differential phase shift ( $\Delta\Phi$ ) and rain rate. The topic explores the use of GNSS-PRO observations, providing valuable insights for the community and demonstrating the potential of this technique for hydrometeor detection and precipitation monitoring.

The work is generally well structured and results are clearly presented. It should be suitable for publication after minor revisions.

To further improve the manuscript, the following points are suggested:

- (1) A short paragraph discussing the motivation for the Yunyao mission in the context of previous PRO missions, such as PAZ and Spire.
- (2) Provide a clear and concise definition of the polarimetric observable  $\Delta\Phi$  at the beginning of the manuscript.
- (3) Expand the Conclusions section with a brief discussion of the potential applications of PRO measurements within the atmospheric science community.

**Specific comments**

**L40.** I think you should rephrase and define the polarimetric observable differential phase shift or at least said what it represents (point (1) of general comments).

Also, I would not say that PRO observable reflects rain rate. The differential phase shift does not reflect a direct measurement on rain rate, it is influenced by the presence of non-spherical raindrops so I would say something more like it reflects the integrated scattering properties of the hydrometeors along the propagation path.

**L138.** You mention the horizontal and vertical polarizations, but I did not see any mention to PRO employing this kind of polarization, instead of RHCP, before that.

You introduce it at section 2.3 but maybe you could mention it briefly before that. Maybe put Eq 12 at the beginning of the article.

**L264.** Cloud itself is not a hydrometeor category; rather, cloud water is. It was demonstrated the sensitivity of PRO to oblate non-spherical raindrops typically present during heavy precipitation events, as well as to oriented frozen hydrometeors, such as snow aggregates. However, given that PRO operates at L-band frequencies, it is unlikely that the technique is sensitive enough to detect small cloud water droplets.

**Figure 10.** Specifiv what the solid and dashed lines represent in the caption.

**L355.** Why the signal decreases near the surface?

**Figure 12.** Could you explain why the heaviest rain rate profile of Figure 12 (b) (dark red color) is superposed with the second heaviest rain rate profile?

**Technical corrections**

**L12.** low-Earth-orbit, while on **L49** it is written as low Earth orbit, without the middle dash.

**L67.** You wrote co-located but then in caption Figure 4 you said collocated.

L71. You wrote POR instead of PRO.

**Figure 2.** You missed a markpoint after Figure 2

**Figure 5.** Missing caption for (c).

**Figure 6.** Missing markpoint after Figure 6

**Table 1.** Missing markpoint after Table 1

**Table 2.** Missing markpoint after Table 2

L200. Geometric optics (GO) was already defined as GO at L189

**L344.** Instead of Fig 11 and Fig 12, you wrote Fig 10 and Fig 11.

**Figure 14.** Units of IMERG precipitation in the x-axis?

---

## Referee Comment (RC3)

On-Orbit Calibration and Performance Validation of the Yunyao Polarimetric Radio Occultation System

Authors: Liang Kan, Fenghui Li, Naifeng Fu, Yan Cheng, Sai Xia, Bobo Xu

**Overall Comments**

In general, this a comprehensive summary of the analysis from the first of the Yunyao polarimetric RO systems. I have several comments below mainly to clarify several points and make this easier for the readers to comprehend.

**Minor Comments**

Title: You mention "on-orbit calibration", but then in the text the calibration is mentioned a couple times but not described / shown in detail.

Line 40. Usually, the abstract is written such that it can be provided "separate" or standalone from the main manuscript. So, don't assume that acronym definitions made in the abstract will be carried over to the main manuscript. Therefore, define  $\Delta \phi$  here and explain what it measures. To help explain to the reader, you can also refer to the Turk et al 2024 manuscript which has a short easy to understand summary of the polarimetric RO concept.

Lines 79-81: It looks like the PAZ group is associated with the ESA - Spire (-Spanish) team, from the way that I read this. It could look like the PAZ group participated in the development of the Spire satellites. Also, you cite the PAZ group validation paper when talking about PlanetiQ.

Line 72. When talking about traditional thermodynamics not being degraded, the correct reference should be Talpe et al. 2025.

Line 125. Define BDS and other GNSS constellations (or make a list of acronyms at the end of the manuscript.

Near Line 180. Can you (in a few sentences) explain what the difference is between open-loop and closed-loop processing? Readers may not be familiar.

Line 185. Suggest you better highlight that the Yunyao constellation (when it is fully deployed) will consist of a mixture of geostationary and polar satellites. I think this is similar to how the

local Indian GNSS system (IRNSS, otherwise known as NavIC) is configured. But your constellation will be global.

Page 115. 53000 profiles from which constellations?

Line 250. Equations 13 and 14: Is (13) for OL and (14) for CL? Clarify this.

Line 260-265. There are two places discussed where 1-sec averaging is being done to produce  $\Delta \phi$  (first one) the "primary PRO observable" (second one). Clarify the need to do two 1-sec averages.

Line 300. You state, "Compared to H- and V-polarized observations, the synthesized data exhibit a higher signal-to-noise ratio and a greater success rate in retrieval." If I interpret this properly, you are saying that a PRO receiver, which splits the received signal into two orthogonal processing chains, does not "degrade" the performance of the usual/traditional non-polarimetric receiver. This is important to better highlight, as it addresses one of the concerns of the user community- That is, that PRO does not compromise or otherwise degrade the use of the usual bending angle data in numerical weather prediction data assimilation systems. Results like yours would better alleviate this concern.

General, End of Section 3. This is an interesting analysis, sort of along the lines of what Cardellach et. al. showed in their series of manuscripts, which you cite. But it would have been much more revealing if you had done the same analysis with PAZ data and then compared the two results. Obviously, PAZ collects less PRO and the analysis periods for the two could not be the same. But PAZ is also in a polar orbit. While your local crossing time is different than the PAZ early morning, over a long analysis period, you may see similar overall statistics as you show in your Tables 3 and 4.

---

## Author Response (AR3)

**For anonymous Referee #1**, we sincerely thank the reviewer for their valuable comments and constructive suggestions. We have carefully addressed each of the points raised. Below, we provide a point-by-point response to the specific comments:

**(1) Regarding the suggestion to enhance the contextual motivation:**

We have incorporated additional content to better situate our work within the current industrial and scientific context. The revised text now reads:

"At present, Chinese commercial aerospace enterprises are actively laying out in this field. Yunyao Aerospace Technology Co., Ltd. is committed to integrating polarized occultation technology into its high-timeliness meteorological constellation. The point is to make up for the deficiency of traditional occultation observations in detecting the microphysical processes of water vapor and precipitation. By capturing $\Delta\Phi$ caused by aspherical hydrometeors within the cloud and rain area, PRO can significantly enhance the constraints on water vapor condensation and phase transition paths during severe convective weather processes, thereby improving the accuracy of numerical models in short-term and imminent precipitation prediction."

**(2) Regarding the definition of $\Delta\Phi$ in the abstract:**

We have clarified the definition of the differential phase at the beginning of the abstract as follows:

"The differential phase ($\Delta\Phi$) is the cumulative phase shift between horizontal and vertical polarizations observed from PRO caused by aspherical hydrometeors along the propagation path, typically measured in millimeters."

**(3) Regarding the expansion of atmospheric science applications in the conclusion:**

We have expanded the conclusion to better reflect the broader potential of PRO in atmospheric science:

"Beyond precipitation detection, PRO observations have broader potential in atmospheric science. The sensitivity of $\Delta\Phi$ to aspherical hydrometeors enables its use in discriminating precipitation types and in identifying mixed-phase and ice-dominated cloud regions. Combined with conventional RO profiles, PRO can constrain not only thermodynamic structures but also microphysical processes aloft, providing a pathway to improve cloud parameterizations in weather and climate models. Furthermore, the high spatiotemporal sampling of PRO constellations supports the analysis of moist processes in data assimilation systems, potentially enhancing the accuracy of short-term precipitation forecasts and the representation of latent heating in tropical cyclones and mesoscale convective systems. As a cost-effective extension of existing RO infrastructure, PRO is poised to bridge gaps between thermodynamic sounding and precipitation observation, advancing the integrated profiling of the moist

atmosphere."

Responses to Specific Comments:

1. (L40) – Definition and explanation of differential phase in the abstract

We have redefined the differential phase in the abstract, specifying that it is derived from the difference between the H- and V-polarized signals obtained from PRO observations. The differential phase directly reflects the scattering characteristics of hydrometeors along the propagation path and, through statistical analysis, can indirectly indicate rainfall rate.

2. (L138) – Definition of differential phase in the introduction

Similarly, the introduction now clearly defines the differential phase, emphasizing that both H- and V-polarized observations are acquired from the PRO technique.

3. (L264) – Attribution of phase difference to hydrometeors

The relevant sentence has been revised to explicitly state that the differential phase is induced by hydrometeors.

4. (Fig. 10) – Addition of relevant content

We have added appropriate content and labels related to Figure 10 as suggested.

5. (L355) – Attenuation of differential phase near the surface

The attenuation of the differential phase near the surface occurs when the radio ray traverses an insufficient thickness of the precipitation layer, leading to a reduced integrated phase shift. Although radio occultation signals near the surface are often affected by multipath interference and low signal-to-noise ratio—which can degrade data quality—the segment in question has undergone quality control and truncation. This behavior is not typical in a general sense; rather, it depends on the actual penetration depth of the ray through the precipitating region.

6. (Fig. 12) – Ray profiles corresponding to the highest and second-highest differential phase values

The ray profile with the highest differential phase and the one with the second-highest correspond to

the ray paths whose tangent points are located at approximately 3 km and 2 km, respectively. This is influenced by the relative motion and geometry between the GNSS satellite and the low-Earth orbit receiver. Although these two rays are close in their tangent altitudes, they are distinct and not superimposed.

Technical Revisions

All technical issues and methodological points raised have been addressed and corresponding modifications have been made throughout the manuscript.

**For anonymous Referee #2**, we sincerely thank the reviewer for their valuable comments and constructive suggestions. We have carefully addressed each of the points raised. Below, we provide a point-by-point response to the specific comments:

(1)

We fully endorse your proposal regarding data sharing, an area in which PAZ has already set a commendable precedent. However, due to the Chinese government's stringent regulations on sensitive data, the commercial nature of this satellite mission, and the fact that its funding is entirely derived from profit-oriented entities, we are currently able to share only the data corresponding to the observation period discussed in this paper (such as Level 2 products). We will actively explore possibilities to make data from other periods and of higher processing levels accessible to the scientific community in the future.

(2)

The study primarily focuses on YunYao PRO data. Numerous existing studies have already validated the correlation between PAZ data and precipitation. We recognize that if YunYao data were collocated not only with PAZ data but also with GPM precipitation data, the final matched YunYao PRO dataset would be insufficient in volume for meaningful research.

(3)

Based on my understanding, the distribution pattern can be attributed to the following factors. The satellite, which operates in a 535 km altitude orbit with an inclination of 97.4° and a local time of approximately 4–6 AM/PM, is equipped with a dual-polarization occultation antenna only on its backward-facing side. Due to the non-overlapping paths of the ascending and descending orbits, combined with the higher signal-to-noise ratio required for polarized occultation measurements, occultation events tend to be preferentially concentrated in the backward direction during both ascending and descending orbital phases.

(4)

In Section 2.2, precise orbit determination, excess phase processing, and profile inversion are also essential steps in the PRO data processing chain. Precise orbit determination requires different parameters to adapt different orbits and satellites. The excess phase processing and profile inversion are indispensable components of PRO data analysis. On one hand, PRO data can be processed through the standard radio occultation procedure to generate relevant products. On the other hand, the process of

obtaining excess phase for both H and V polarizations in PRO is consistent with standard RO processing.

(5)

The satellite mission in question was conducted as a technology demonstration payload. Building upon our existing, standardized occultation platform, we implemented a moderate upgrade to validate our capability in tracking and processing polarized radio occultation signals. Accordingly, the forward-facing antenna, signal channels, and processing methods retained the conventional occultation configuration. In contrast, the backward-facing antenna, channels, and data processing procedures were specifically designed to accommodate the characteristics of polarimetric occultation—such as the handling of dual-polarization data, fusion of polarized signals, integration with standard occultation retrieval workflows, and calculation of differential phase.

The transition between open-loop and closed-loop tracking was set at a tangent height of 0 km, following expert recommendations presented at the ROMEX meeting. The resulting measurement performance on highest vs lowest tracking altitudes is illustrated in Figures 10 and 13, which we deem sufficient for presentation without introducing additional diagrams.

Furthermore, the signal-to-noise ratio (SNR) characteristics—for horizontal (H) and vertical (V) polarizations, as well as the combined signal—were found to be consistent with those observed by payloads such as PAZ, SPIRE, and PlantiQ. Specifically, the variation trends of SNR (H, V) closely resemble those of the combined SNR, with magnitudes approximately equal to $1/\sqrt{2}$ of the latter. Given this alignment with established missions, we have chosen not to include further comparative analysis in this regard, and instead focus our discussion on the correlation experiment between PRO (Polarimetric Radio Occultation) profiles and precipitation.

(6)

The primary reason for setting delta alpha corrected as constant below 20 km is the frequent risk of L2 signal loss of lock in this region, coupled with the generally compromised reliability of the signal even before loss of lock occurs. Therefore, it is standard practice to use L2 and L1 combinations from above 20 km to correct for ionospheric errors. Additionally, below this altitude, the ionospheric influence becomes relatively small in magnitude compared to the dominant bending effect caused by atmospheric refraction. This approach is consistently adopted in several established processing software packages, such as ROPP and ROAM.

(7)

We have added the following formulas in the manuscript to illustrate the combined signals:

The H- and V-channel I/Q streams are time- and frequency-aligned and coherently combined into a complex baseband as followed:

$$I_H + jQ_H \cdot conj(I_V + jQ_V) = (I_{H-V} + jQ_{H-V}), \tag{12}$$

$$\Delta\varphi_{H-V} = \angle(I_{H-V} + jQ_{H-V}), \tag{13}$$

$$S_{combined} = (I_H + jQ_H) + (I_V + jQ_V) \cdot e^{-j\Delta\varphi}, \tag{14}$$

where $I_H$ and $I_V$ represents horizontal and vertical polarization in-phase component, respectively; $Q_H$ and $Q_V$ represents horizontal and vertical polarization orthogonal component, respectively; $\Delta\varphi_{H-V}$ represents the relative phase difference between horizontal and vertical polarized signals; $S_{combined}$ represents the combined complex signal.

(8)

In the study by Katona et al., ray paths below 6 km or 12 km were used to construct a circle with a diameter of 2° or 0.6°, and the average GPM precipitation data within this circle was taken as the reference precipitation for the corresponding PRO event. In this study, since YunYao data has a sampling rate of 100 Hz, which is higher than that of PAZ, we utilize the GPM precipitation data corresponding to each sampling point along the actual ray path. Although this approach may consumes more resources and time in practice, we believe it provides a more accurate validation of the precipitation sensitivity of YunYao data.

(9)

Your point is very insightful. Indeed, below 5 km, the penetration depth of BDS PRO in YunYao data is not as deep as that of GPS and GLONASS. This may be attributed to the frequency of the BDS B3 signal. The B3 signal operates at a higher frequency, which is more susceptible to atmospheric refraction effects and signal attenuation when propagating through dense atmospheric layers.

(10)

We will include more detailed descriptions in the captions of Figures 11 and 12.

(11)

For Figures 11 and 12, we have added matched infrared brightness temperature data along the PRO profiles and marked the freezing level in the $\Delta\phi$ profiles to better observe the transitions between ice and liquid water. In order to represent the actual ray paths more accurately, the bending angle has been taken into account in the ray tracing for these figures, although its impact is minimal in practice.

(12)

At altitudes of 5–6 km, clouds are often concentrated, and the PRO rays traverse the longest path within this region. Since $\Delta\phi$ represents the accumulated phase difference along the ray path, water contents, if present in a PRO event, tends to maximize its contribution to $\Delta\phi$ around 5–6 km.

(13)

The term "stable" here refers to the stability of the occultation retrieval results. Specifically, it indicates that the temperature and pressure profiles are smooth and continuous in the vertical direction, without significant discontinuities or spikes. The errors in the retrieval results are bounded and predictable, and their magnitude and variation patterns are consistent with theoretical expectations.

**For anonymous Referee #3**, we thank the reviewer for their valuable comments and constructive suggestions. We have carefully considered each point raised and have made corresponding revisions to the manuscript. Below, we provide some response to the specific comments:

(1)

The on-orbit calibration of Yunyao data serves to verify its usability, refine the data processing methodology, and explore potential application value. Since payload characteristics vary across satellite platforms, systematic biases and error sources during operation may differ. Therefore, on-orbit calibration acts as an end-to-end validation of the entire chain from data acquisition to product generation.

(2)

In the main text, we have revised relevant definitions by incorporating the terminology introduced by Turk et al. (2024) regarding PRO and $\Delta\varphi$. This inclusion helps clarify these concepts for readers.

(3)

"Line 72. When talking about traditional thermodynamics not being degraded, the correct

reference should be Talpe et al. 2025."

"Line 125. Define BDS and other GNSS constellations (or make a list of acronyms at the end of

the manuscript."

Thanks. The corresponding parts in the article have been modified.

(4)

Near Line 180, we have added a technical explanation of open-loop and closed-loop tracking modes to improve comprehension for readers:

"Specifically, closed-loop tracking employs a feedback-controlled architecture where the receiver continuously adjusts its carrier and code replicas based on discriminator outputs from received signals. This phase-locked loop mechanism enables real-time compensation for Doppler shifts and dynamic trajectory variations, providing optimal tracking precision under stable signal conditions. In contrast, open-loop tracking operates without phase feedback, instead utilizing predicted ephemeris, atmospheric models, and pre-calibrated trajectory data to directly generate local signal replicas at anticipated signal phases. This forward-prediction approach eliminates loop latency to rapid signal dynamics, making it particularly robust in the lower troposphere where refraction and intense multipath frequently cause deep signal fades and phase discontinuities."

(5)

The Yunyao constellation is dedicated to meteorological sounding and is designed with both Sun-synchronous and low-inclination orbits. Due to operational constraints, geostationary orbits are not employed. Sun-synchronous orbits enable globally distributed radio occultation events with high temporal resolution, while low-inclination orbits enhance RO coverage in mid- and low-latitude regions. It should be noted that RO satellites operate passively—they receive rather than transmit signals—which distinguishes them from active navigation systems such as IRNSS.

(6)

Page 115. 53000 profiles from which constellations?

Thank you for raising this question. The 53,000 profiles represent the aggregate number of radio occultation events collected by the on-board satellites of Yunyao Meteorological Constellation from all four global GNSS constellations: GPS (USA), GLONASS (Russia), Galileo (EU), and BeiDou (China).

(6)

"Line 250. Equations 13 and 14: Is (13) for OL and (14) for CL? Clarify this."

"Line 260-265. There are two places discussed where 1-sec averaging is being done to produce

$\Delta\Phi$ (first one) the "primary PRO observable" (second one). Clarify the need to do two 1-sec

averages."

The formula of data processing about OL and CL data have been clarified.

The original description in Lines 260–265 was unclear. We have revised this paragraph to clarify that the $\Delta\varphi$ requires only a 1-s smoothing step.

"Because the $\Delta\Phi$ originates from hydrometeors (rain, cloud, ice crystals), we set $\Delta\Phi$ to zero at 30 km under water-free conditions to remove a profile-wide offset; all subsequent $\Delta\Phi$ values are referenced to this level. The calibrated phase is smoothed with a 1-s filter and linearly detrended along the full profile to produce the polPhs file (Padullés et al., 2024). In parallel, the excess phase from the synthesized signal undergoes standard RO retrieval to derive dry and wet profiles, which are interpolated to a 0.1-km grid to form the resPrf file (Padullés et al., 2024) for collocation of $\Delta\Phi$ with thermodynamic fields. For calibration, IMERG-identified rain-free events are used to derive the in-orbit antenna pattern and to remove any residual ionospheric imprint on $\Delta\Phi$ (Padullés et al., 2020). The result of calibrated and smoothed $\Delta\Phi$ is the primary PRO observable. After these steps, remaining $\Delta\Phi$ variability can be attributed to differences in H- and V-component propagation induced by non-spherical, preferentially

oriented hydrometeors along the path. Fig. 9 illustrates the smoothing and detrending: in (a) the light-blue curve is $\Delta\Phi$ after de-slipping, the blue curve is the 1-s smoothed $\Delta\Phi$, and the red dashed line is the linear trend; in (b) the detrended $\Delta\Phi$ after smoothing is shown."

(7)

You state, "Compared to H- and V-polarized observations, the synthesized data exhibit a higher signal-to-noise ratio and a greater success rate in retrieval." If I interpret this properly, you are saying that a PRO receiver, which splits the received signal into two orthogonal processing chains, does not "degrade" the performance of the usual/traditional non-polarimetric receiver. This is important to better highlight, as it addresses one of the concerns of the user community- That is, that PRO does not compromise or otherwise degrade the use of the usual bending angle data in numerical weather prediction data assimilation systems. Results like yours would better alleviate this concern.

We thank the reviewer for this comment and for having accurately captured the central argument of our paper. Our results demonstrate that the PRO technique is not a trade-off but rather a seamless extension of traditional RO capabilities: it adds new polarimetric observables while maintaining the quality of traditional bending angle data and introducing new data products such as polarimetric phase differences. We fully agree that clearly articulating this "win-win" situation is essential to alleviate user concerns and foster its adoption.

(8)

At the end of Section 3, the result of analysis is similar to Paz. The study primarily focuses on YunYao PRO data. Given the well-established correlation between PAZ PRO data and precipitation demonstrated in existing literature, this study focuses specifically on the analysis of the Yunyao PRO dataset. We recognize that if YunYao data were collocated not only with PAZ data but also with GPM precipitation data, the final matched YunYao PRO dataset would be insufficient in volume for meaningful research. A comparative analysis of the precipitation sensitivity between Yunyao and PAZ data under similar spatiotemporal conditions will yield more meaningful insights once a substantial volume of Yunyao observations is accumulated following an extended period of on-orbit operation.

**For anonymous Referee #2**, i apologize for the lack of detail in my previous responses to some of the questions. I will now address your comments more thoroughly, with particular focus on the points you raised in comments 9 and 12:

(9)

Thanks for your remind. While the raw electromagnetic propagation through a neutral troposphere at GNSS L-band exhibits negligible frequency dispersion, different GNSS signals may still show different loss-of-lock behaviors due to receiver tracking loop sensitivity and signal structure differences.

Modern GNSS carrier and code tracking loops (PLL/DLL) rely on signal power, modulation format, and the presence of a pilot channel to maintain lock. In some signal designs (e.g., earlier BDS B1/B3), the pilot carrier may be absent or defined differently, which forces the carrier tracking loop to rely solely on the data channel with frequent navigation bit transitions. In contrast, signals with an independent pilot (such as modern L1C/B1C) generally enable higher tracking stability and more robust phase/frequency estimation under the same conditions.

These implementation and signal design differences can result in higher loss-of-lock rates on certain frequencies, even though the tropospheric refractive delay itself is essentially nondispersive across the L-band and thus does not intrinsically cause larger tropospheric bending or attenuation differences between B1 and B3. Any measurable differences in refractivity estimates may instead be associated with residual ionospheric effects, receiver tracking performance under multipath/noise, or structural differences in the signal and receiver algorithms.

(12)

Here, "optical path" is used to indicate the actual ray path length through hydrometeor-rich/cloudy regions that can introduce additional attenuation or phase/angle perturbations.

According to standard cloud classification, mid-level cloud types such as altocumulus and altostratus typically occur in the mid-troposphere, ranging roughly from a few kilo meters up to around 6–8 km depending on weather regime and latitude, whereas low clouds (e.g., cumulus, stratocumulus) are typically below ~2–3 km and high clouds (e.g., cirrus) occur above ~6 km. This classification reflects the relative vertical occurrence of cloud types, not a single altitude of "maximum cloud concentration".

I have revised the sentence to:

*"In the mid-troposphere,a few kilometers in altitude, certain cloud layers such as altocumulus and altostratus are commonly present. Radio occultation rays that intersect these layers may traverse an*

*extended optical path through moist/cloudy regions, potentially influencing attenuation or refractivity*

*retrievals depending on local water/ice content and particle phase distribution."*